# Soil Type Affects Organic Acid Production and Phosphorus Solubilization Efficiency Mediated by Several Native Fungal Strains from Mexico

**DOI:** 10.3390/microorganisms8091337

**Published:** 2020-09-02

**Authors:** Dorcas Zúñiga-Silgado, Julio C. Rivera-Leyva, Jeffrey J. Coleman, Ayixon Sánchez-Reyez, Susana Valencia-Díaz, Mario Serrano, Luz E. de-Bashan, Jorge L. Folch-Mallol

**Affiliations:** 1Facultad de Arquitectura e Ingeniería, Institución Universitaria Colegio Mayor de Antioquia, Carrera. 78 N° 65-46 Robledo, Medellín 050034, Colombia; dorcas.zuniga@colmayor.edu.co; 2Centro de Investigación en Biotecnología, Universidad Autónoma del Estado de Morelos, Cuernavaca 62209, Morelos, Mexico; susana.valencia@uaem.mx; 3Facultad de Farmacia, Universidad Autónoma del Estado de Morelos, Av. Universidad No. 1001, Col. Chamilpa, Cuernavaca 62209, Morelos, Mexico; julio.rivera@uaem.mx; 4Department of Entomology and Plant Pathology, Auburn University, Rouse Life Sciences Building, Auburn, AL 36849, USA; jjc0032@auburn.edu (J.J.C.); luz@bashanfoundation.org (L.E.d.-B.); 5Cátedras CONACyT, Institute of Biotechnology, Universidad Nacional Autónoma de México, Ave. Universidad 2001, Col. Chamilpa, Cuernavaca 62209, Morelos, Mexico; ayixon.sanchez@mail.ibt.unam.mx; 6Centro de Ciencias Genómicas, Universidad Nacional Autónoma de México, Av. Universidad 2001, Col. Chamilpa, Cuernavaca 62209, Morelos, Mexico; serrano@ccg.unam.mx; 7Environmental Microbiology Group, Northwestern Center for Biological Research (CIBNOR), Av. IPN 195, La Paz 23096, Baja California Sur, Mexico; 8Bashan Institute of Sciences, 1730 Post Oak Ct, Auburn, AL 36830, USA

**Keywords:** biosolubilization of phosphorus, soil sorption capacity, organic acids, *Trichoderma*, *Aspergillus*

## Abstract

Phosphorus (P) is considered a scarce macronutrient for plants in most tropical soils. The application of rock phosphate (RP) has been used to fertilize crops, but the amount of P released is not always at a necessary level for the plant. An alternative to this problem is the use of Phosphorus Solubilizing Microorganisms (PSM) to release P from chemically unavailable forms. This study compared the P sorption capacity of soils (the ability to retain P, making it unavailable for the plant) and the profile of organic acids (OA) produced by fungal isolates and the in vitro solubilization efficiency of RP. *Trichoderma* and *Aspergillus* strains were assessed in media with or without RP and different soils (Andisol, Alfisol, Vertisol). The type and amount of OA and the amount of soluble P were quantified, and according to our data, under the conditions tested, significant differences were observed in the OA profiles and the amount of soluble P present in the different soils. The efficiency to solubilize RP lies in the release of OAs with low acidity constants independent of the concentration at which they are released. It is proposed that the main mechanism of RP dissolution is the production of OAs.

## 1. Introduction

Phosphorus (P) is considered a macronutrient with minor mobility and availability for plants in most tropical soils [1], thus being one of the most serious factors limiting plant growth. The low mobility of bioavailable phosphate ions (H_2_PO_4_^−^ and HPO_4_^2−^) is due to their retention by the colloidal mineral constituents of the soil [2], which determines that only a small proportion of the available ions is present in the soil solution [3]. In general, the highly weathered soils of the tropics stand out for their high capacity to fix P on the surface of secondary minerals (gibbsite, ferrihydrite, goethite) [4]. Furthermore, the association of P with free Fe (OH)_2_ and Al (OH)_3_ ions in the soil solution, make P very insoluble since it forms stable complexes, particularly at a pH < 5 [5] or complexes with Ca^2+^ ions at pH > 6.5 [6,7]. The application of rock phosphate (RP) has been extensively studied as a method to cope with the needs of the phosphoric requirements of crops [4,8]. However, the amount of available P released to the soil from the direct application of RP is too low to satisfy the demand of the crops [9,10], limiting its widespread use. Besides, there is currently a situation with RP exhaustion, making the full use of this resource essential [7]. Faced with this situation, the idea arises to implement low technological cost alternatives to improve the acquisition of P by plants; these include the use of rhizosphere and endophytic microorganisms that promote the solubilization of P from chemically unavailable forms such as RP [5,11].

The beneficial effects of these organisms have been widely reported but frequently yield inconsistent results [12,13], possibly because the sole inoculation of microorganisms into the soil-plant system alters the relationship among the microorganisms inhabiting the microhabitat of the rhizosphere ecosystem [14,15]. It is also assumed that the differential response depends on the plant species [16], on the mineralogical composition of the soil [17,18], on the interactions between distinct groups of soils, and the different populations of microbes, their competition and their metabolism in response to these factors [19,20]. Also, the type and amount of inoculum that affects microbial cell physiology [21,22], as well as the availability of carbon substrates as energy sources [9,23], culture conditions, and environmental conditions [5,24,25], among other variables. Among the soil-borne microorganisms, certain filamentous fungi have a ten-fold greater ability to solubilize P than many rhizobacteria [26]. There are two main mechanisms that are involved in P solubilization by microbes (although other less studied possibilities arise from acidification due to microbial respiration, redox activity of microorganisms due to a wide range of extruded secondary metabolites, sink theory, etc.). One is the production of microbial enzymes during the decomposition of organic matter [3,27], and the other, the secretion of organic acids (OAs) that dissolve RP and whose organic anions compete with phosphate ions for adsorption sites on the surface of soil clays [26,28]. These acids are products of microbial metabolism, mostly through oxidative respiration or by fermentation of carbonaceous substrates (e.g., glucose) [29]. Although the secretion capacity and quality of OAs is basically determined by gene regulation, it can also be affected by proton excretion in the assimilation of NH_4_^+^ [30], which affects both the type of organic acid(s) secreted and the desorption of P ions from adsorption sites, including mechanisms such as the chelation of Al and Fe [31].

Although it is true that the solubilization capacity of P is associated with the decrease of pH due to the release of OAs, the effect of the soil mineralogy on the secretion profile of these acids and the solubilization efficiency of P has not been documented. Therefore, our hypothesis is that the mineralogy of the soil and attributes of the fungal isolate will affect the secretion profile of OAs and that this modulates the efficiency of in vitro RP microbial solubilization. Thus, the objective of this research was to evaluate the effect of the P sorptive capacity of soil on the secretion profile of OAs, and its effect on the in vitro solubilization efficiency of RP with different fungal isolates.

## 2. Materials and Methods

### 2.1. Sampling Area

Samples for fungal isolation were collected from three locations in the state of Morelos-México and corresponded to Horizon A (0 to 20 cm) of an Andisol (Tres Marías—Municipality of Huitzilac; 19°02′18′′ N 99°15′11′′ W; 2803 m above sea level [masl]), an Alfisol (Experimental Center for Agronomic Sciences FCA-UAEM; 18°58′35′′ N 99°13′36′′ W, 1824 masl), and a Vertisol (Cuentepec—Municipality of Temixco 18°51′36′′ N 99°19′29′′ W; 1480 masl). The physicochemical composition of soils is shown in Appendix A.

### 2.2. Isolation of Fungal Strains

Isolation of fungi was performed from as previously described [32] from three different locations described above. To try to avoid variation as much as possible, we collected the samples from three “milpas”, which contained the same plants, but in different soils (as depicted in Appendix A). Milpas are agricultural units that contain in the same area of land and intermingled, beans (*Phaseolus vulgaris* L.), corn (*Zea mays* L.), chili (*Capsicum annum* L.), and tomato (*Solanum lycopersicum* L.), so in the different locations the same plant species were co-cultivated. Rhizosphere, rhizoplane, and endophytes from beans (*P. vulgaris* L.), corn (*Z. mays* L.), chili (*C. annum* L.) and tomato (*S. lycopersicum* L.) from each “milpa” were considered. Five samples from each plant were collected from each of the three different locations (180 samples in total: three kinds of soils, four plants species, three microhabitats, five samples per plant).

For the isolation of rhizosphere fungi, collected roots still covered with soil from each site were immediately immersed in 10 mL of sterile 0.85% NaCl solution and centrifuged for 10 min at 1000× *g*. Serial dilutions were made from the supernatant, and 0.5 mL of each dilution was spread on Potato Dextrose Agar (PDA, Sigma-Aldrich Química, S.L.Toluca, Mexico) medium (originally developed by Koch in 1881, but cited in reference [33] or Bengal Rose Agar (BRA-Sigma-Aldrich Química, S.L.Toluca, Mexico) medium [34] both media contained streptomycin sulfate (300 mg L^−1^) and chloramphenicol (100 mg L^−1^) (Sigma) to inhibit bacterial growth. The dishes were incubated in complete darkness for five days at 25 ± 1 °C. The high osmotic pressure of BRA inhibits the development of fast-growing fungi, such as strains belonging to *Rhizopus* and *Mucor* genera, and allows the outgrowth of strains of *Trichoderma* and *Aspergillus,* among others [35].

To isolate rhizoplane fungi, 1 g of roots from each sample was transferred to a 100 mL Erlenmeyer flask. The roots were washed three times with a sterile 0.85% NaCl solution. Finally, the roots were submerged in 20 mL of water with 10% peptone. The roots were shaken for 5 min at 100 rpm and followed by 5 min of sonication. Serial dilutions were made from the supernatant, and 0.5 mL of each dilution was spread on PDA or BRA medium as described above.

To isolate endophytic fungi, the same roots used to isolate rhizoplane fungi were separated from the supernatant. These roots were surface sterilized in 3 cycles of the following: cycle one: the roots were submerged in 5% NaClO for 1 min with constant agitation, washed with sterile water and immersed in 70% ethanol for 30 s. After these treatments, the roots were washed three consecutive times with sterile water. Cycle two: the roots were submerged in 2% NaClO for 1 min with constant stirring, washed with sterile water, and immersed in 70% ethanol for 20 s. The roots were washed three consecutive times with sterile water. Cycle three: the roots were submerged in 0.5% NaClO for 1 min with constant stirring, washed with sterile water, and immersed in 70% ethanol for 10 s. The roots were washed three consecutive times with sterile water. After the three cycles, the roots were sonicated for 5 min in a 1% Tween 80 (Sigma) diluted solution. Finally, the roots were dried on sterile absorbent paper.

From this treatment, half of the roots were dissected into 1 mm segments while the other half of root samples were transferred to a mortar for maceration. Both the roots segments and the root maceration mixture were spread on separate Petri dishes with PDA or BRA medium in complete darkness, for five days at 25 ± 1 °C. A total of 360 treatments were obtained three soils, four plants species, five samples per plant, three microhabitats per plant (rhizosphere, rhizoplane, and endophytic), and two culture media), and three Petri dishes were spread for each one. From these treatments, 31 fungal morphotypes were isolated and conducted to axenic cultures.

### 2.3. Analysis of Rock Phosphate Solubilization

Preliminary analysis to select strains with high RP solubilization capacity were performed using solid MOH medium: [36] (in g/L) (1.0 NaCl; 0.2 CaCl_2_·2H_2_O; 0.4 MgSO_4_·7H_2_O; 1.0 NH_4_NO_3_; 10 L^−1^ glucose; 3.5 g unacidulated RP (Fosforita 28P^®^, which is composed by 8% (Ca_5_(PO_4_)_3_, 38% CaO, 14% SiO_2_, 3% F, 1% C, 0.50% Al_2_O_3_, 0.40%, Fe_2_O_3_, 0.10% MgO 0.30% SO_4_, 10% Na_2_O, 0.10% K_2_)_._ Three Petri dishes were spread for each individual isolate and incubated for five days at 25 °C ± 1. Growth rate and formation of solubilization halos were determined.

### 2.4. Molecular and Phylogenetic Analysis

From each of the four selected isolates, spore suspensions were prepared with 1 mL of 0.01 M CaCl_2_·2H_2_O at a density of 1 × 10^6^ spores per mL. Erlenmeyer flasks with 75 mL of PDB liquid medium were inoculated with one mL of each spore suspension. After five days of growth at 25 °C ± 1 and shaken at 100 rpm, the mycelia of the four strains were harvested for genomic DNA extraction performed with a DNAeasy Plant Mini Kit (Qiagen Inc., Valencia, CA, USA) using the manufacturer’s protocol. DNA was stored at −20 °C prior to use. The internal transcribed spacer region (ITS) was amplified from each of the genomic DNA samples using a semi-nested PCR protocol and the ITS1 universal primers (5ʹ-TCCGTAGGTGAACCTGCGG-3ʹ) ITS4 (5ʹ-TCCTCCGCTTATTGATATGC-3ʹ). PCR amplification was performed using the thermocycler (Bio-Rad S1000, Hercules, CA, USA) with the following cycle parameters; the initial denaturation step at 94 °C for 5 min, 30 cycles: denaturation at 94 °C for 1 min, annealing at 52 °C for 30 s, extension at 72 °C for 1.5 min, and a final extension step at 72 °C for 10 min. The amplicons obtained from the ITS rDNA region were sequenced by the Sanger method.

The obtained nucleotide sequences were analyzed and compared with closely related sequences and aligned with those deposited in the GenBank database using BLAST (https://blast.ncbi.nlm.nih.gov/Blast.cgi). To do this, the sequences of the ITS region from the three *Trichoderma* and the *Aspergillus* strains were compared against the NCBI nucleotide collection (nt) database (limiting the alignment only with sequences of confirmed type material) to select phylogenetic neighbors for each strain.

For each taxon, 50 blast hits were recruited; all sequences were clustered with the CD-HIT at 99% identity [37]. The grouped sequences were aligned with the MAFFT software version 7.310 (March 2017) for the Linux system with default settings [38]. Gblocks version 0.91b (January 2002) was used to remove misaligned regions of DNA alignment, while less stringent selection was activated (allows for smaller space, end blocks, and less stringent flanking positions) [39]. The resulting cured alignment was used in FastTree software version 2.1.10; option: fasttree-gtr (generalized reversible time model) < nucleotide_alignment > finaltree [40]. Other options were set as default. Figtree version 1.4.3 was used to show the summarized and annotated tree produced by the steep reconstruction. General clades were grouped to achieve a better final graphic presentation.

### 2.5. RP Solubilization in Liquid Medium

To evaluate the effect of soil mineralogy on the profile of the organic acids (OAs) and the solubilization of the RP, the four monosporic cultures of the above selected fungi were evaluated according to Osorio and Habte [36]. Thus, 75 mL of MOH medium were supplemented with 3.5 g/L of unacidulated RP and 3.5 g/L of each soil (Andisol, Alfisol and Vertisol) for each treatment. The pH was adjusted to 7.0 with 0.1 M NaOH and autoclaved. The treatments were carried out in 250 mL Erlenmeyer flasks. Spore suspensions for each isolate were prepared with 1 mL of 0.01 M CaCl_2_·2H_2_O at a density of 1 × 10^6^ spores per mL. Prior to in vitro inoculation, the viability of the fungal spores was verified, according to Calish et al. [41]. One mL of each suspension was inoculated in each Erlenmeyer flask. A control without inoculation (WI) received 1 mL of the 0.01M CaCl_2_·2H_2_O solution. The cultures were incubated at 25 ± 1 °C on an orbital shaker at 100 rpm for six days. Each treatment had four replicates.

Subsequently, the cultures were vacuum filtered through a cellulose filter and supernatants were collected in 50 mL conical tubes to measure the pH (Conductronic PC45, Mexico) and the soluble P concentration using the molybdate blue method at 890 nm in a spectrophotometer (Genesys 20, ThermoSpectronic, Philadelphia, PA, USA) [42]. After filtration, each sample was centrifuged for 10 min at 5000× *g* to separate the fungal cells and particulate material. Still, the supernatant was vacuum filtered in duplicate through 0.45 µm Ø and 0.2 µm Ø Millipore filter paper, respectively. To collect the fungal biomass (in mg) the MOH liquid medium was transferred to a 100 mL Erlenmeyer with progressive heating from 27 to 31 °C, 100 rpm for 1 h. Then, the samples were kept at 31 °C in an oven. The hot liquid medium was transferred to conical tubes for centrifugation for 5 min at 1000× *g* at 31 °C. Subsequently, it was vacuum filtered through a cellulose filter. The biomass was washed several times and collected in a Petri dish. Finally, it was dried in an oven at 60 °C for 24 h and then weighed.

### 2.6. Organic Acid Analysis Using HPLC

Crude supernatant filtrates containing the secreted fungal OAs were centrifuged for 15 min at 5000× *g*. The supernatants were then filtered in duplicate through a 0.2 µm filter (Millipore, GTBP, Kenilworth, NJ, USA). Reverse Phase High-Performance Liquid Chromatography (HPLC) was performed on a Hitachi Lachrom Elite instrument (model pump L-2130, automatic injector L-2200, column oven L-2300, UV detector L-2400). The operating conditions consisted of a Mobile Phase A: 0.005M Phosphate buffer (KH_2_PO_4_) pH 7.0; Mobile phase B: Methanol (CH_3_OH) 99% HPLC grade. A flow of 1.2 mL/min per gradient of 30% KH_2_PO_4_ and 70% CH_3_OH with a ramp of 12 min and a total run time of 18 min. The separation of the OAs was carried out on a Zorbax-(NH_2_) column (4.6 × 250 mm, Agilent Technologies Laboratories, Inc. Santa Clara, CA, USA) [43] at a temperature of 30 °C. The injection volume was 20 µL. The retention time (RT) of each signal was recorded at a wavelength of 210 nm. Analytical grade standards (Sigma) of the following OAs were used: citric, glutaric, malic, succinic, oxalic, pyruvic (1 mg/mL), fumaric, and tartaric (100 µg/mL) acids [44]. Each treatment had four analytical replicates. The HPLC results were analyzed with data processing software—EZchrom Elite Ver. 3.1.3. The detected OAs were identified by comparing the peaks of their retention times and the areas under the curve of their chromatograms with the standards.

### 2.7. Experimental Design and Statistical Analysis

To verify our hypothesis, the experimental design was completely random. The treatments had a 4 × 5 factorial arrangement where the soil factor had four levels (Andisol, Alfisol, Vertisol, Without Soil), and the fungal isolate factor with five levels (strain BMH-0059, strain BMH-0060, strain BMH-0061, strain BMH-0062, and the control Without Inoculation (WI)). Each treatment had four replicates and two repetitions. In each experimental unit, the P concentration was measured in the solution (mg L^−1^), pH, and seven OAs were identified and quantified. Analyses of variance (ANOVA) and multiple comparison tests (LSD All-Pairwise Comparisons Test) were performed to compare mean values for the different factors and variables analyzed (*p* < 0.05) using Statistic 8.

The analysis of OAs by HPLC was carried out with a General Linear Model (GLM) [45], which models the effect of soil mineralogy, strain type, and their interactions on the P solubilization efficiency, controlling the secretion profile of OAs. The latter as covariates of the concentration of P. To reduce the number of covariates and achieve covariates independent of each other, a principal component analysis (PCA) was performed considering as variables pyruvic, fumaric, tartaric, succinic, malic, oxalic, and citric acids. OAs were integrated through a PCA [46], and in the GLM, they used the first two principal components as covariates [46]. The PCA analysis was based on the correlation matrix for each principal component. Those OAs that contributed at least 10% of the variance in the normalized length of the principal component and that at the same time had an *r* ≧ [0.7] were considered significant. In the case of significant differences in the GLM, the Tukey test (*p* ≤ 0.05) was used. All the analyzes were carried out with the GM statistical software, the generic R version 3.4.3. (R Core Team 2017), and the PCA with the package ade4 version 1.7-15.

## 3. Results

### 3.1. Isolation and Selection of Fungal Strains

From the 360 treatments described previously, a total of 31 fungal morphotypes were isolated from the selected plant species in different soil types. Preliminary tests indicated that of these, ten isolates showed the best efficiency to acidify the medium and solubilize RP (data not shown). Based on several preliminary evaluated variables (growth rate, size of solubilization halos, biocompatibility among the ten strains, etc.; data not shown), four of them, morphologically consistent with the genera *Trichoderma* (three endophytes: one from corn/alfisol, one from chili/alfisol; and one from chili/vertisol) and *Aspergillus* (one from the rhizosphere of the bean/andisol condition), were selected for further biochemical, molecular, and phylogenetic analyzes.

### 3.2. Molecular and Phylogenetic Analysis

The molecular identification of the four fungal isolates were confirmed by sequencing of the internal transcribed spacer region, and the phylogenetic reconstruction was performed with reference sequences and with the Maximum Likelihood algorithm (Appendix A). BMH-0059 (isolated from corn/alfisol) was consistent with *Trichoderma crassum* with high branching support and 100% sequence identity (Figure 1a). BMH-0060 (isolated from bean/andisol) is phenotypically and molecularly consistent with the *Aspergillus* clade, it groups near the *A. niger* and *A. awamori* subclades, with real branching divisions that accept values with 92% identity for the reference sequences. However, strain BMH-0060 was placed in its own branch with good support value. Based on the morphological observations and its position in the phylogeny, we propose that this strain is a novel species closely related to *A. awamori* (Figure 1b). BMH-0061 (isolated from chili/alfisol) is also consistent with the *Trichoderma* genus but grouped into a separate branch, being its closest relatives’ species in the *koreanum* subclade with a branch support of 89%. This value is much lower than for the other strains. We propose that this may be a new undescribed species of *Trichoderma*, which for the purposes in this investigation, will be considered as a *Trichoderma* sp. (Figure 1c). Strains encoded as BFV761 (isolated from chili/alfisol), and BMH-0062 (isolated from chili/vertisol), share 98.36% sequence identity, also consistent with the *Trichoderma* group, and both closely related to *Trichoderma pubescens* (BMH-0061) but in its own distinctive branch with a 100% support value. We propose that the strain selected for further analyses (BMH-0062) is also a novel variety related to *T. pubescens* (Figure 1d) due to its unique, robust position in the phylogenetic reconstruction.

### 3.3. RP Solubilization in the Liquid Medium

The interaction between soil and each fungal isolate was evaluated to determine its effect on the secretion profile of OAs and the in vitro dissolution of RP. On the sixth day of incubation in MOH medium with unacidulated RP, each of the isolates (BMH-0059, BMH-0060, BMH-0061, and BMH-0062), solubilized different amounts of P (measured as H_2_PO_4_^−^ mg L^−1^) according to the presence or absence of the three different soils. It should be noted that in all the treatments, the pH value was inversely proportional to the amount of P in solution, and the interaction between these factors was significant (*p* ≤ 0.01) (Figure 2).

In the absence of soil, the control treatment without inoculation (WI), presented a low concentration of P in a solution of 2.8 mg L^−1^ and pH of 6.8 (Figure 2a,b). However, when different soils and isolates were added, the concentration of P significantly changed. For example, when BMH-0059 was inoculated in the medium without soil, the soluble P concentration was 12.19 mg L^−1^ and a pH of 4.0. Nonetheless, when the soil component was added to the system, the concentration of P in the solution decreased significantly (*p* ≤ 0.05) in all the treatments.

In Andisol, BMH-0059 generated significantly lower concentrations of soluble P (*p* ≤ 0.01) (4.6 mg L^−1^ at pH of 6.8, Figure 2b). Isolates BMH-0060, BMH-0061 and BMH-0062 performed much better, the trend was maintained with soluble P concentrations up to 17.5 times more than the control, these being 22.07 mg L^−1^ at pH 5.7; 29.18 mg L^−1^ at pH 4.0, and 35.21 mg L^−1^ at pH 2.6, respectively (Figure 2).

In Alfisol, BMH-0059 performed poorly, presenting soluble P concentrations similar to those observed in the control without soil and inoculation (2.14 mg L^−1^, although the pH was lower than the control: 4.0; Figure 2a,b). Isolates BMH-0060, BMH-0061 and BMH-0062 performed well, showing soluble P concentrations up to 11.3 times more than the control, 24.18 mg L^−1^ and pH 3.3; 34.65 mg L^−1^, and pH 2.8 and 27.69 mg L^−1^ and pH 3.2, respectively.

In Vertisol, isolate BMH-0059 showed its best performance leading to concentrations of P in the solution of 8.6 mg L^−1^ and pH of 6.6 (Figure 2a,b). This soil was also where the highest soluble *p* values were obtained for isolates BMH-0060 and BMH-0062, with concentrations up to 33 times more than the control, 31.21 mg L^−1^ and pH 3.1; and 36.36 mg L^−1^ and pH 2.9, respectively. Isolate BMH-0061 was more versatile, performing as well as in Alfisol (34.23 mg L^−1^ and pH 3.3) (Figure 2).

The relative efficiency of RP dissolution regardless the strains used but in accordance with the soil mineralogy was Vertisol > Alfisol > Andisol. The isolates solubilized, on average, 20.6 times more P than the control treatment.

### 3.4. Profile of OAs and Its Correlation with the P Concentration in Different Soil Types

Figure 3 shows evidence of a differential secretion profile of OAs according to the type of soil and strain. The chromatographic profile showed significant differences (*p* ≤ 0.05) for the OAs secreted by BMH-0059 in each of the soils. The concentration of P in solution was low, ranging from approximately 4 to 8 µg/mL for Andisol, Alfisol, and Vertisol (Figure 2). It was observed that all the acids tested were secreted in every soil type (Figure 3). However, in treatment WS, the concentration of P in solution was 12,191 ± 0.569 µg/L (Figure 2 and Appendix A), where only fumaric, succinic, and oxalic acids were detected and in very low concentrations (Figure 3d).

BMH-0060 presented significant differences of soluble P (*p* ≤ 0.01) among soils. Thus, in Andisol, Alfisol and Vertisol the concentration of P in solution was 22.072 ± 2.73, 24.185 ± 0.385, and 31.215 ± 0.351 µg/mL, respectively (Figure 2). In Alfisol (Figure 3b), the fungus secreted high concentrations of pyruvic acid compared to the control (4.126 ± 0.026 µg/mL) (Appendix A) and did not secrete tartaric or succinic acids. While in Vertisol, the concentration of tartaric acid was the highest achieved for any of the OAs in any condition (5.765 ± 0.219 µg/mL) (Figure 3c and Appendix A). In contrast, the treatment WS (Figure 3d) showed a concentration of P in solution of 31.188 ± 0.240 µg/mL (Appendix A), while in this condition, BMH-0060 secreted only fumaric, succinic, malic, and oxalic acids at very low concentrations (Figure 3).

Strain BMH-0061 also presented highly significant differences of soluble P (*p* ≤ 0.01) among soils. Thus, for Andisol, Alfisol and Vertisol the concentration of P in solution was 29.187 ± 0.400, 34.652 ± 0.255 and 34.240 ± 0.209 µg/mL respectively (Figure 2 and Appendix A), whereas the treatment WS, showed the highest concentration of soluble P (40.615 ± 0.883 µg/mL) (Appendix A) in relation to the rest of the treatments. This strain secreted fumaric, malic, and oxalic acids at very low concentrations in all the treatments (Figure 3).

Finally, strain BMH-0062 also showed highly significant differences of soluble P (*p* ≤ 0.01) among soils. Thus, for Andisol, the concentration of P in solution was 35.217 ± 0.0800 µg/mL (Figure 2 and Appendix A), for Alfisol 27.693 ± 0.714 µg/mL (Figure 2 and Appendix A), and for Vertisol 36.369 ± 0.711 µg/mL (Figure 2 and Appendix A). In Andisol (Figure 3a), no citric acid production was detected, but pyruvic, tartaric, and succinic acids were present in significant amounts. In Alfisol (Figure 3b), only a high concentration of tartaric acid was detected (3.126 ± 0.221 µg/mL) (Figure 3), while the rest of the measured acids showed low concentrations. In Vertisol (Figure 3c), tartaric acid was the only relevant OA. In the treatment WS, this strain showed the second-highest concentration of P in solution (37.505 ± 0.350 µg/mL) (Appendix A) with respect to the rest of the treatments, although the fungus secreted very low concentrations of fumaric, succinic, malic and oxalic acids (Figure 3d).

### 3.5. Interaction between Soil Mineralogy and OA

Considering the premise that the mineralogy of the soil could be affecting the secretion profile of OAs (types and quantity), and this would influence the microbial solubilization efficiency of P, a PCA was performed. Based on the coordinates of the main factors that provided the greatest variance, a general linear model was performed that analyzed the effect of the type of soil (Andisol, Alfisol, Vertisol, Without Soil (WS)), and the isolates BMH-0059, BMH-0060, BMH-0061, and BMH-0062) on the concentration of P. The coordinates of the mentioned factors generated in the PC analysis were considered as covariates. In Andisol, Vertisol, and treatments Without Soil, a compact and similar distribution of microorganisms was observed (Figure 4). Within Alfisol, BMH-0059 and BMH-0062 grouped together, while BMH-0060 and BMH-0062 formed separate individual groups (Figure 4). Based on the GLM model, there is no effect type of soil that correlates the kind and amount of acid with the concentration of P in solution.

Regarding PCA for media inoculated without soil, (Figure 5), two components explained 51.74% of the total variance. PC1 with 31.32% and PC2 with 20.42%. PC 1 separates oxalic, malic, and succinic acids from tartaric, fumaric, citric, and pyruvic acids. It is observed that the pyruvic and citric acids have the highest correlations with -0.93, and their contribution percentage exceeded 10%. In PC 2, tartaric acid showed the highest correlation with 0.75 and a contribution of 39.27% (Figure 5, Appendix A).

Soil type, microorganisms, and their interaction affected the phosphorus concentration. However, the acids grouped in the space defined by PC1 and PC2 axes, did not show significant differences (Appendix A). The control treatment (Without Soil) had the highest concentration of P (mean ± SE; 22.16 ± 3.14 µg/mL; Tukey test, *p* < 0.05), followed by Vertisol (27.62 ± 2.86 µg/mL), while there were no significant statistical differences between Alfisol and Andisol, with 22.16 ± 3.14 µg/mL and 22.77 ± 2.96 µg/mL, respectively. On the other hand, BMH-0059 presented the lowest concentration of P in solution (6.90 ± 1.00 µg/mL), BMH-0060 an intermediate level (27.16 ± 1.06 µg/mL), while BMH-0061 and BMH-0062 contributed with the highest concentration of P in solution (*p* < 0.05; 34.67 ± 1.07 µg/mL, and 34.19 ± 1.02 µg/mL). The treatment BMH-0061 Without Soil, showed the highest concentration of P in solution, and the lowest concentration was achieved when BMH-0059 was inoculated together with Alfisol (Figure 6).

## 4. Discussion

The results in this work confirm the hypothesis that the mineralogy of the soil and the fungal isolates present affect the secretion profile of OAs and that this modulated the efficiency of RP solubilization. It was particularly important to properly characterize the fungal isolates by morphological, biochemical, molecular, and phylogenetic methods to select those of interest for further studies and development since frequently microorganisms that exhibit a great potential to solubilize P, turn out to be plant pathogens for agriculturally important crops [47].

Potentially, the selected microorganisms may serve as promising clean agroinputs (these are ubiquitous fungi and have not been genetically modified) for the inoculation and production of biofertilizers and phytostimulants that could be available relatively soon, after some additional ongoing investigation. The classification of the fungal isolates depicted here was performed with phylogenetical inference methods that are very powerful because, contrary to the widely used Neighbor-Joining algorithm that primarily considers the number of differences, the Maximum Likelihood algorithm was used with a cured alignment of sequences. This sequence selection and the optimization criteria used by the phylogenetic reconstruction algorithm gives very reliable results that are strongly supported. Based on the robust phylogenetic reconstruction carried out, the proposal of a new strain of *Trichoderma* sp. (*T. morelensis*) is presented, although additional markers need to be explored to determine if strain BMH-0061 indeed is a new *Trichoderma* species. Isolate BMH00-60 could also be a new species since it stands alone in its own branch with a high support value, but additional molecular markers will need to be investigated to explore this possibility since Aspergilli can often be difficult to classify. In addition, the evaluated species have been reported as friendly to the environment and plant health [48,49,50]. The species of the genera reported here are strongly involved in the cycling of nutrients, as well as in the transformation of P through solubilization, mineralization, and immobilization. The effectiveness of these microorganisms could result in an expected reduction in the use of agrochemicals in agroproductive systems [51]. This work reports for the first time the effect of soil mineralogy and fungal isolates on the secretion profile of OAs and its relationship to the solubilization efficiency of RP by three novel native species of fungi (three endophytic *Trichoderma,* and one rhizosphere *Aspergillus* sp.), isolated from different Mexican soils.

These four fungi were selected based on their prominent ability to solubilize inorganic P (in the form of RP) by showing potential to improve the release of the element over a short time frame in vitro. In this initial screening, the fungal isolates that showed the highest capacity to solubilize RP compared to other native strains included in this study were in the following order: BMH-0061 > BMH-0062 > BMH-0060 > BMH-0059 (data not shown). The solubilization of P was favored by the reduction of the pH, which was attributed to the diffusion of several OAs secreted by these fungi. In the soil, one advantage of a microorganism secreting OAs over inorganic acids (IAs) is that OAs produce fewer negative effects on the quality of the soil, including the structure, compaction, or moisture retention, among others. Furthermore, the release of more H^+^ ions due to their low acidity constants (at the same pH as IAs) are determining factors in their efficiency [26].

To investigate the mechanisms of acidulation, three factors should be considered: the pH values, the types, and amounts of OAs, and the feedback on environmental acidity [26]. Despite these premises, the results of successful isolates in vitro are often inconsistent when evaluated in the field [12]. One important factor that is omitted in in vitro studies is the mineralogy of the soil, which determines the efficiency of P microbial solubilization [28,31]. In this research, it was shown that soil mineralogy influenced the efficiency of the strains to secrete OAs and solubilize RP. This was consistent with Andisol showing a lower concentration of P in solution in relation to the other treatments since it is composed, predominantly by Allophane clays with a strong retaining P capacity. In this soil type, all the fungi secreted a mixture of OAs to acidify the medium and to release P in solution. The low availability of phosphate ions (H_2_PO_4_^−^ and HPO_4_^2–^) in Andisol was probably due to strong retention by the colloidal mineral constituents of the soil [2], that ultimately results in only a small fraction of P was found in solution [3]. In general, the highly weathered soils of the tropics such as the Andisols, Alfisols, and Vertisols stand out for their high capacity to fix P on the surface of secondary minerals (Allophane, Kaolinite, Montmorillonite, Gibbsite, Ferrihydrite, Goethite) [4], and this aspect limits the efficiency of P microbial solubilization. The search for microorganisms that retain their P solubilization efficiency when inoculated into these soils becomes a challenge for any investigation because the rapidly solubilized P could precipitate with free Fe (OH)_2_ and Al (OH)_3_, if present in the soil solution, causing P to form very insoluble and stable complexes particularly at pH < 5 [5]. Additionally, P can precipitate with Ca^2+^ ions at pH > 6.5 [6,7].

For the first time, we report data about how the mineralogy of the soil and the type of fungal isolate influence the secretion profile of fungal OAs, and demonstrate that the P solubilization efficiency by a microorganism is not necessarily related to the type or amount of released OA [52]; rather, although releasing fewer OAs microorganisms are more efficient to solubilize P when they secrete OAs with low acidity constants [26]. For example, in the soil treatments, BMH-0061 and BMH-0062 solubilize higher amounts of P than the other strains despite a low concentration of secreted organic acids. P solubilization in the presence of soils was independent of the concentration in which the OAs were in solution. In our study, a series of seven OAs with different acidity constants, determined the ability of four fungal strains to change the acidity of the culture medium in the presence of soil, affecting the solubilization of P. The acidity constants for the OAs analyzed in this research are: pyruvic: Kα = 2.8 × 10^−3^, fumaric: Kα = 8.8 × 10^−4^, tartaric: Kα = 9.2 × 10^−4^, succinic: Kα = 6.2 × 10^−5^, malic: Kα = 3.5 × 10^−4^, oxalic: Kα = 5.6 × 10^−2^, and citric: Kα = 7.5 × 10^−4^. Oxalic acid was the major component that contributes to the acidity in the medium, has a first-degree ionization constant of approximately ~ 10 to 1000 times higher than that of the other OAs evaluated [26].

Our findings agree with those of Li et al. [26], who, in their studies with *Aspergillus* and *Penicillum*, reported oxalic acid as the most influential OA in the efficiency of P solubilization. Likewise, Alam et al. [53], reported that oxalic and citric acids had been the most common OAs produced by microorganisms in the corn rhizosphere. Additionally, Akintoku et al. [54] further support our findings where they reported that acids such as lactic, succinic, gluconic, and fumaric were released when using tricalcium phosphate with different species of fungi. Usually, fungi are better P solubilizers than bacteria [26]. This is probably due to the kind of organic acids that bacteria secrete which have higher acidity constants than the ones found in this study (for example succinic and gluconic acid [55] compared to oxalic acid found here).

While acidity has a significant influence on the P solubilizing capacity of microorganisms, it is not the most important variable that determines the solubilization efficiency of P. In this respect, we concur with Li et al. [26], who in their studies with *A. niger* and *P. oxalicum*, found that the growth of these fungi was relatively good in acidic environments (pH = 3~4) compared to other fungi and bacteria. However, under conditions of extreme acidity (pH > 3), only the biomass of the fungi increased, and their citric acid secretion began to dominate the secretion of other OAs. Particularly in our study, citric acid was present in the treatments, although in very low concentrations; furthermore, in treatment WS, citric acid was not detected. Presumably, citric acid is derived from the Krebs cycle, so its synthesis and secretion entail a very high energy expenditure for the cell. It also has a very high acidity constant, so the cell could refrain from its release until only absolutely necessary. It is likely that the cell preferably releases other OAs with low acidity constants and at low concentrations, preserving the metabolic efficiency of the organism. Such is the case of treatments where no soil was used, where citric acid was not released but the secretion of oxalic acid prevailed. It is known that when glucose is metabolized, citric acid is produced, and by decreasing the pH value to ~2.0, the cells produce an inactive aconite hydratase blocking the TCA cycle and inhibiting the release of OAs with negative consequences for the solubilization of P [56,57]. Consequently, very low pH values are counterproductive in preserving the microbial solubilization efficiency of P.

## 5. Conclusions

To the best of our knowledge, this is the first time that the effect of soil mineralogy and type of fungal isolate on the secretion profile of OAs, and its relationship to the solubilization efficiency of RP has been reported. Four novel species of efficient P solubilizing fungi were isolated from different soil types and were characterized as *Trichoderma* (3) and *Aspergillus* sp. (1), being isolated as endophytes, the former and rhizosphere as the latter. HPLC analysis revealed that the soil type and fungal isolate determined the secretion profile of OAs and the solubilization of P. The efficiency to solubilize RP lies in the release of OAs with low acidity constants independently of the concentration in which they are released. Therefore, for this investigation, oxalic acid dominated the acidity in the medium with a first-degree ionization constant approximately ~100 times higher than fumaric and succinic acids, where these three organic acids appear to be the main determining factors for P solubilization.

## Figures and Tables

**Figure 1 microorganisms-08-01337-f001:**
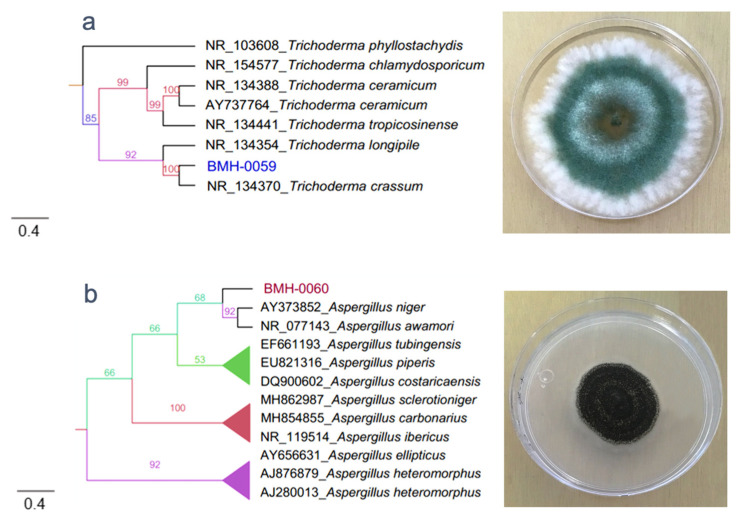
Phylogenetic tree generated by FastTree analysis using a MAFFT alignment of ITS2-4 nucleotide sequences obtained from the type material. The tree includes strains related to (**a**) Strain BMH-0059 (left) is identified as *T. crassum*. (**b**) Strain BMH-0060 (left) is identified as related to *A. awamori*. (**c**) Strain BMH-0061 (left) is identified as *Trichoderma* sp. (**d**) Strain BMH-0062 (left) is identified as closely related to *T. pubescens*. Bootstrap values (>50%) are labeled in color on the branch nodes. A representative image of the colony is shown on the right side of each alignment. The bar indicates 4 percent of changes per site.

**Figure 2 microorganisms-08-01337-f002:**
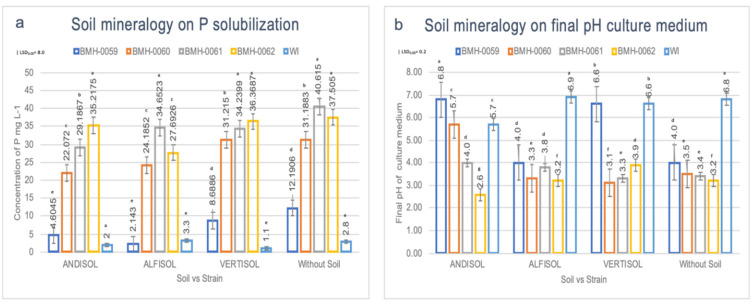
Evaluation of the effect on P solubilization in different soils inoculated with different fungal isolates. ANOVA multiple comparison tests (**a**) soluble P (H_2_PO_4_^−^) (| LSD_0.05_ = 8.0), and (**b**) final pH of the culture medium. (| LSD_0.05_ = 0.2). Three soils with different mineralogy were evaluated: Andisol, Alfisol, and Vertisol, and a control Without Soil (SI) with four different fungal strains Bars represent the standard error. All values shown are the average of four replicates per treatment (*p* ≤ 0.01).

**Figure 3 microorganisms-08-01337-f003:**
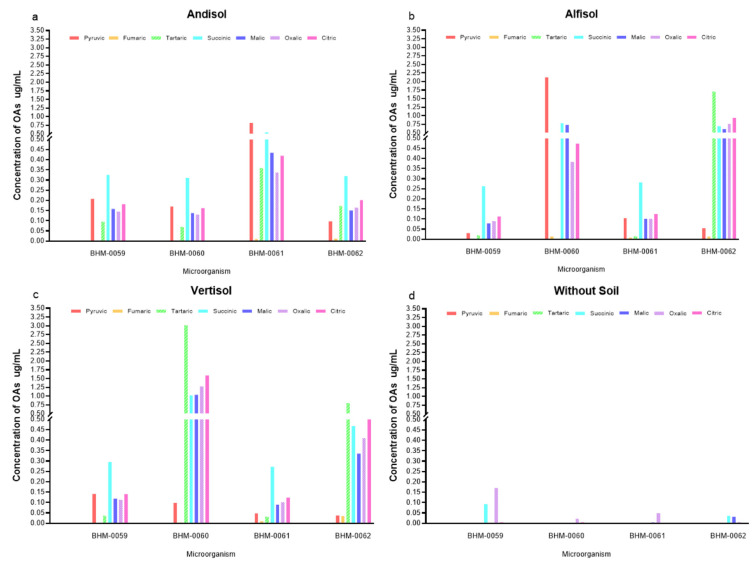
Profile of OAs produced by different fungi when incubated in different soils. Three types of soils were evaluated: (**a**) Andisol, (**b**) Alfisol, (**c**) Vertisol, and (**d**) a control Without Soil (WS). Strains BMH-0059, BMH-0060, BMH-0061, BMH-0062 were tested. All values shown are the average of four replicates per treatment. (*p* ≤ 0.01).

**Figure 4 microorganisms-08-01337-f004:**
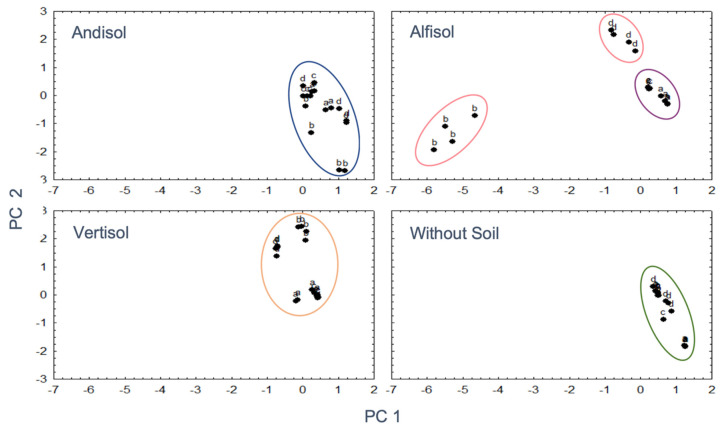
Distribution patterns of fungal isolates in the four different soil types. It was observed that except in Alfisol, the fungal isolates did not display much variability between them, which results in a strong grouping. Ellipses demonstrate microorganism community groups a: BMH-0059, b: BMH-0060, c: BMH-0061, d: BMH-0062.

**Figure 5 microorganisms-08-01337-f005:**
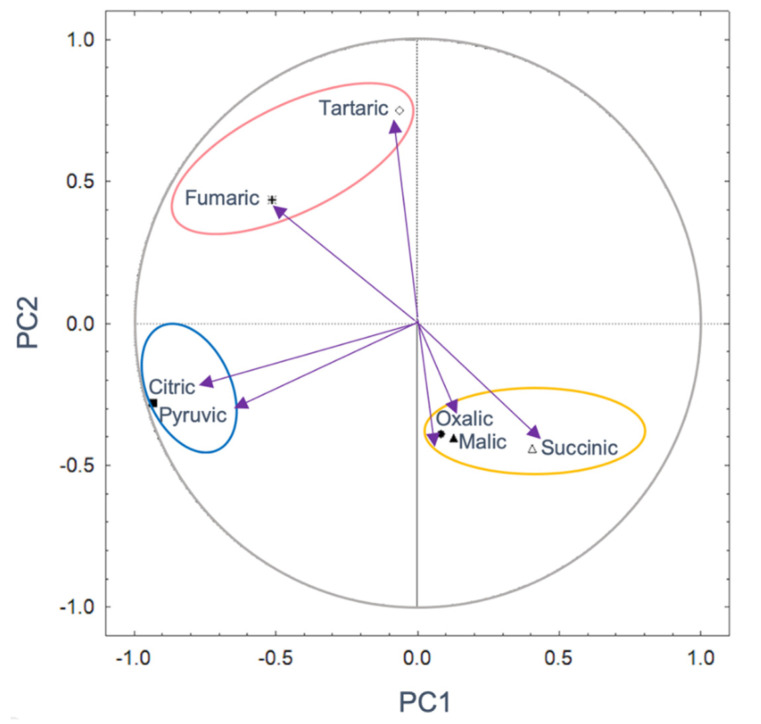
Principal components based on the correlation of the kind and amount of organic acid without considering the soil mineralogy. PC1 31.32%; PC2 20.42%. Significant parameters are indicated by continuous lines (*p* < 0.05). The ellipses show groups of communities of OAs.

**Figure 6 microorganisms-08-01337-f006:**
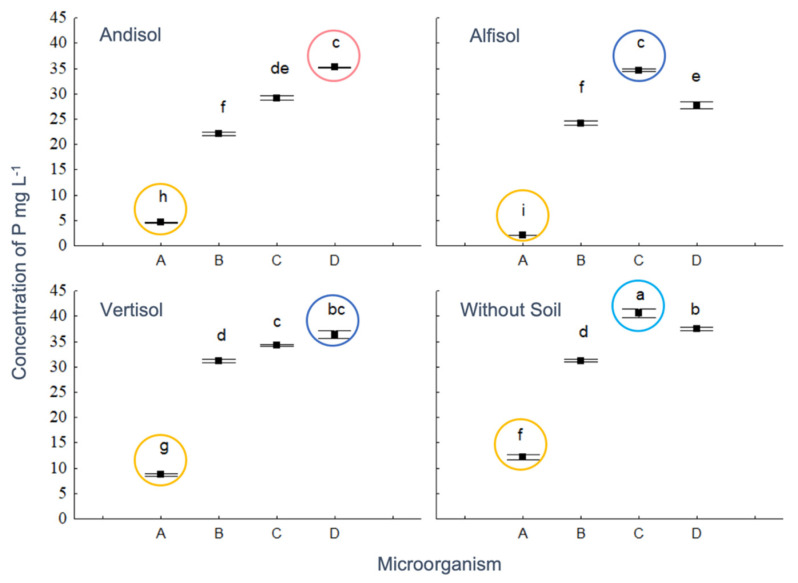
Effect of the interaction of soil and fungal isolate on P content, having PC1 and PC2 as covariates, which were not significant. Circles indicate the microorganisms in the x-axis: A: BMH-0059, B: BMH-0060, C: BMH-0061, D: BMH-0062. Different letters denote statistically significant differences (*p* < 0.05).

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
