# Peer review of "Soil Type Affects Organic Acid Production and Phosphorus Solubilization Efficiency Mediated by Several Native Fungal Strains from Mexico"

_microorganisms, 2020, doi:10.3390/microorganisms8091337_

Round 1

Reviewer 1 Report

The English of the manuscript is generally fine however there are several periods that are too long. Please try to avoid too long phrases that may compromise clarity.

Abstract

Line 29: Please, explicit the acronym PSM since it’s the first time you mention it.

Line 31: Please, add the acronym “OA” between brackets after “organic acids”.

Line 32: Please, add “Aspergillus strains”

Line 35: please specify that according to your data in these conditions OA are the main responsible...

Introduction

Line 48: correct spelling is gibbsite

Line 49: “free Fe (OH) and Al (OH) ions”, please, indicate these chemical species in the correct ionic form.

Line 52-54: It is suggested to mention that currently there is a crisis due to rock phosphates exhaustion, considering this problem makes even more crucial to fully exploit RP added to the soil

Line 55: in literature not only rhizospheric microorganisms have been reported…

Line 58: you address the inconsistent results to the modification of the microhabitat but it could also be that the lack of results is caused by the competition that the inoculum faces

Line 59-60: please replace “alters the dynamic…” with “alters the relationships among the microorganisms inhabiting the rhizospheric microhabitat”

Line 64: “availability of (the) carbon substrate (s) as energy source(s)” why between brackets?! Please clarify

Please clarify what do mean with “Microculture conditions”. The last 3 lines of this period could be better explained.

Line 67: this is made possible both by… why are considered only lytic enzymes produced during decomposition of organic matter and not more generally fungal enzymes? Also, inorganic material can be altered by fungal enzymes e.g. phosphatases

Line 71-75: apparently only OAs production and proton extrusion are considered, other suggested mechanisms of P solubilization are reported in literature (e.g. acidification due to microbial respiration, redox activity of microorganisms due to a wide range of extruded secondary metabolites, sink theory)

Materials and methods

84: please replace with “Sampling area”

92-96: the experimental design for what concerns the sampling is not explained clearly. Please specify step by step both the number of samples and variables

103: I suggest to replace with “…the development of fast-growing fungi, such as strains belonging to Rhizopus and Mucor genera allowing the development of strains of Trichoderma …”

97, 105 and 110: Please add at least one reference for each of isolation method used.

124: if as mentioned above you sampled 5 samples per plant the number of treatments doesn’t match. With the term environment do you mean rhizosphere, rhizoplane and inner tissues of roots? Please substitute it with “microhabitat”. However, considering you isolated separately from roots segments and maceration mixture shouldn’t the treatments be multiplied for 4? Please clarify.

128-129: please report the brand of the chemicals used. You mention solid medium, so please report the agar concentration?

Section 2.3 Biocompatibility among strains: it is not clear what’s the purpose of this test considering that you further test only single strains and not combinations, in addition you’re not showing the results of this test but you address to use it to select the species to test further. If this section doesn’t provide useful information for this study I suggest to remove it

135: which type of barriers? What do you think these may represents? Biocompatibility does not show up only in this form…

138-140: please explain better the method and furthermore this statement on the selection of the species is incongruent with line 233-236

172: what do mean with “monosporic strains”? did you meant monosporic cultures of the strains. Please describe more clearly what you have done

228: “ade4” is a package not a library0, which version?

Results

Line231-233: please replace “various crop” with “selected plant species”. 31 fungal morphotypes corresponding to how many isolates? These 10 isolates to which species/morphotypes belong to?

233-236: if 10 isolates showed the best efficiency how did you select these 4 on which criteria? Which variables were considered?

3.2 Molecular identification

Throughout the section there are incongruences between what you point out as identified and -related and proposal of new species. For example about BMH-0060 at first you collocate it related to niger and awamori, then you define it as awamori and in the end you propose it as new species, furthermore in the figure caption you claim it as identified as awamori. Same issues are present also for the others isolates and the incongruences are throughout the manuscript. Is suggested to make a proposal of identification in this section and further report only the code in following sections.

For the Trichoderma genus are required up to 6 markers (Samuels an Hebbar 2015) including the translation elongation factor 1, with only ITS1/4 it is not enough to claim it to be a new species.

Line 251: “so there is no clear species context for this strain according to these results” this doesn’t add any information and it’s not currently clear

Line 265: Please, correct “crissum” with “crassum".

Line 293: “generating concentrations of P” please replace generating with “leading to”

Line 298-299: please replace “the strain used” with “the used strains”

Line 300: please remove all the phrase after treatment. It doesn’t add any information and creates confusion on the experimental design.

342: Figure 3.d is wrong it seems you entered the results of the same species 4 times looking at the results reported in the text only two strain produce the 4 acids while the other two showed different secretion of 3 acids. Please correct the figure

347-355: this part is not suitable for the results section, maybe you could rephrase it and add it to materials and methods

375: please add type after soil

Line 376: “the acids grouped in PC1 and PC2” substitute with “grouped in the space defined by PC1 and PC2 axes”. did not play a role? Do you mean that it’s not statistically different? That’s a little bit different.

4.Discussion

The discussion you propose is quite wide and in some parts resemble an extension/commentary of the results section. Please reconsider this section to make it more robust and avoid redundancies of contents.

394-397: I suggest to remove the part of phrase after “development” conserving the context of the species studied In the manuscript.

398-399: this statement is pointless, doesn’t add anything to the discussion

412-413: see comment 394

450-451: this study’s results do not support this statement

456: considering you didn’t perform enzymatic assays how could you just address the efficacy to OAs quantity and quality.

459-461: this statement is not supported and is not related to the discussion of your results

474-475: this phrase is out of purpose for discussing your results on fungi

References

608: correct “Penicillium oxalicum and and Aspergillus Niger”

Reviewer 2 Report

The authors have prepared the very interesting paper that studies the effect of various fungi on the solubility of phosphates and the production of organic acids in the presence of soils. However, the title refers to "soil mineralogy", and in fact the authors do not provide data on the mineral composition of the used soils. Therefore, in my opinion, the title does not correctly reflect the content of the paper. In connection with the possible effects of other soil properties on the processes under study (organic matter content and amount, pH, grain size distribution, cation exchange capacity, etc.), Table S1 should be included in the main body of the paper. In turn, phylogenetic trees can be transferred into supplemental material.

Remarks on the content of the paper:

Lines 7-23. It is necessary to arrange the dots when marking the affiliation.

Line 29. The abbreviation PSM occurs for the first time, but there is no decoding.

Line 30. It is necessary to give an explanation of what the authors mean by "P sorption capacity".

Line 33. Abbreviation OA occurs for the first time, but there is no decoding.

Line 60. What is the "dynamics of the rhizosphere ecosystem". Is it about succession or a change in the composition of the microbial community?

Line 52, 53. The word "satisfy" is repeated in the sentences. It is necessary to find a synonym.

Table S2. What does the "Total" line at the bottom of the table mean? Explain, please.

Lines 92-96. It is necessary to more clearly explain how many samples and where they were taken from. I understand that not all plants have produced effective strains of fungi. Maybe these plants should not be mentioned?

Line 234. Strains of fungi from rhizoplan were not used also. Why mention them then?

For ease of perception, the paper should be slightly reduced.

Round 2

Reviewer 1 Report

Line 443 -"Aspergilli": do not italicize

Author Response

The typo is correct now, I am sorry for this mistake.

We have changed the title according to your suggestion, and we have mad a better explanation on how the samples were taken.